# Sleep restriction impairs visually and memory-guided force control

**Sarah A. Brinkerhoff**[1]*, **Gina M. Mathew**[2], **William M. Murrah**[3], **Anne-Marie Chang**[4], **Jaimie A. Roper**[1], **Kristina A. Neely**[1]

**1** School of Kinesiology, Auburn University, Auburn, Alabama, United States of America, **2** Program in Public Health, Renaissance School of Medicine, Stony Brook University, Stony Brook, NY, United States of America, **3** Department of Educational Foundations, Leadership, and Technology, Auburn University, Auburn, Alabama, United States of America, **4** Department of Biobehavioral Health, College of Health and Human Development, The Pennsylvania State University, University Park, Pennsylvania, United States of America

* sbrinkerhoff@auburn.edu

## Abstract

Sleep loss is a common phenomenon with consequences to physical and mental health. While the effects of sleep restriction on working memory are well documented, it is unknown how sleep restriction affects continuous force control. The purpose of this study was to determine the effects of sleep restriction on visually and memory-guided force production magnitude and variability. We hypothesized that both visually and memory-guided force production would be impaired after sleep restriction. Fourteen men participated in an eleven-day inpatient sleep study and completed a grip force task after two nights of ten hours' time in bed (baseline); four nights of five hours' time in bed (sleep restriction); and one night of ten hours' time in bed (recovery). The force task entailed four 20-second trials of isometric force production with the thumb and index finger targeting 25% of the participant's maximum voluntary contraction. During visually guided trials, participants had continuous visual feedback of their force production. During memory-guided trials, visual feedback was removed for the last 12 seconds of each trial. During both conditions, participants were told to maintain the target force production. After sleep restriction, participants decreased the magnitude of visually guided, but not memory-guided, force production, suggesting that visual attention tasks are more affected by sleep loss than memory-guided tasks. Participants who reported feeling more alert after sleep restriction and recovery sleep produced higher force during memory-guided, but not visually guided, force production, suggesting that the perception of decreased alertness may lead to more attention to the task during memory-guided visual tasks.

## Introduction

It is recommended that adults receive at least seven hours of sleep each night for optimal mental and metabolic health and performance [1]. However, according to the Centers for Disease Control, 35% of adults in the United States achieve fewer than seven hours of continuous sleep

---

**Data Availability Statement:** All relevant data are within the manuscript and its Supporting Information files.

**Funding:** This project was supported by a pilot grant (PI: A-M. Chang) from the Pennsylvania State

University Clinical and Translational Sciences Institute (funded by the National Center for Advancing Translational Sciences, National Institutes of Health (NIH), through Grant UL1TR002014) and institutional funds from the College of Health and Human Development of the Pennsylvania State University (to A-M. Chang and O.M. Buxton). The Research Electronic Data Capture (REDCap) survey platform and the Clinical Research Center are supported by the National Center for Advancing Translational Sciences, NIH, through grants UL1 TR002014 and UL1 TR00045. The content is solely the responsibility of the authors and does not necessarily represent the official views of the NIH. The funders had no role in study design, data collection and analysis, decision to publish, or preparation of the manuscript.

**Competing interests:** The authors have declared that no competing interests exist.

per day, and these individuals are more likely to be obese, physically inactive, and have chronic health conditions such as diabetes, depression, stroke, and coronary heart disease [2]. Therefore, it is important to understand the multi-system effects of restricted sleep schedules on health and functioning.

In addition to the above-mentioned health consequences, sleep restriction negatively affects speed [3–8] and accuracy [7] on tests of sustained attention such as the psychomotor vigilance task (PVT). Performance in more complex domains, such as working memory, is also impaired after sleep restriction [7, 9, 10]. Working memory is the temporary storage and manipulation of information, and is an intersection between memory, perception, and the attentional control of behavior [11], and is necessary for motor skill acquisition [12, 13]. Indeed, the motor system and working memory tasks recruit common neural pathways [14–18]. Considering that sleep restriction negatively affects working memory, it follows that sleep restriction may also affect performance of motor tasks that rely on working memory. However, there is a dearth of research examining such tasks.

Although it is unclear whether sleep restriction affects the performance of continuous motor tasks, prior research demonstrates that sleep restriction impairs performance of discrete button-press tasks. For example, individuals exhibit more variability on the PVT (e.g., standard deviation of reaction time) after total sleep deprivation [19, 20]. However, discrete button-press tasks confound sensory, motor, and cognitive processes into a single dichotomous response. Furthermore, many occupational tasks and activities of daily living require relatively short periods of continuous motor output, such as carrying a cup of hot coffee. Sleep-restricted participants exhibit slower reaction time [21, 22] and variability [22] in driving simulation tasks, but there is a lack of literature on the effects of sleep restriction on tasks requiring both continuous motor output and working memory, such as playing an instrument or participating in an athletic activity.

Therefore, the purpose of this study was to evaluate the effects of sleep restriction on force production during visually and memory-guided force control tasks. We hypothesized that sleep restriction would impact memory-guided but not visually guided force production.

## Methods

### Ethics statement

The Institutional Review Board at Penn State approved all procedures, which were consistent with the Declaration of Helsinki. After receiving a complete description of the study, each participant provided written informed consent prior to enrollment in screening and study procedures. Participants received monetary compensation for their involvement.

### Participants

Healthy young adult male participants, ages 20–35, were recruited for this study through websites (such as Penn State research websites and Craigslist) and flyers posted at the Penn State University Park campus and surrounding areas.

A clinician at the Penn State Clinical Research Center reviewed participants' medical history and conducted physical exams to assess physical health, and a clinical psychologist conducted a structured clinical interview with each participant to determine psychiatric and psychological suitability. Participants were included if they were deemed healthy and were adherent to screening procedures. Women were excluded due to the effects of the menstrual cycle on sleep and circadian rhythms [23] and adults over 35 were excluded due to the effect of aging on sleep patterns [24]. Exclusion criteria were tobacco or drug use (confirmed by urine toxicology), excessive alcohol consumption, prescription medication use, hearing or vision

impairment, neurological disorder, night or shift work within the previous three years, travel across > 2 time zones within the previous 3 months, acute or chronic medical conditions, or history of sleep disorder, which was evaluated through the Sleep Disorders Questionnaire [25]. Exclusion criteria also included metal implants that would be unsafe for magnetic resonance imaging and any of the following sub-clinical metabolic disturbances: body mass index of $\leq$ 18 kg/m$^2$, systolic blood pressure of $\geq$ 130 mmHg or a diastolic blood pressure of $\geq$ 85 mmHg, hemoglobin A1C glycosylation level of $\geq$ 5.7% (pre-diabetic or diabetic per ADA criteria), HDL cholesterol level of < 40 mg/dL, LDL cholesterol level of $\geq$ 145 mg/dL, plasma triglyceride level of $\geq$ 150 mg/dL, plasma triglyceride to HDL cholesterol ratio of $\geq$ 3.5, fasting glucose $\geq$ 100 mg/dL, and/or waist circumference > 102 cm.

## Procedures

As previously described [7, 26], the current work was part of an 11-day inpatient study that followed a within-subjects design. Participants were admitted to the Clinical Research Center of the Pennsylvania State University at approximately 11:00 on admission day to a private, windowless room under constant (artificial) light levels (< 100 lux in the angle of gaze during wake; complete dark at 0 lux during scheduled sleep) and temperature conditions (20˚-22˚C). The private room contained a single bed, a desk for administration of cognitive batteries, and a bathroom with a shower. Participants were not permitted to nap, sit, or recline in bed during scheduled wake times and were monitored by research assistants to confirm adherence. Light-emitting personal devices such as mobile phones and laptop computers were removed 2 hours before scheduled bedtime and were returned at least 2 hours after scheduled wake time to limit exposure to the alerting effects of blue light near the sleep episode [27, 28].

Each participant underwent all three of the following conditions: baseline, sleep restriction, and recovery sleep (Fig 1). The rested baseline condition consisted of three nights of 10 hours in bed. To ensure that participants were adapted to the in-lab baseline condition, participants were instructed to spend 10 hours in bed each night at home for the week prior to admission into the study. In addition, their sleep schedules were monitored using wrist actigraphy and time-stamped calls at bedtimes and wake times during this week. Following the rested baseline

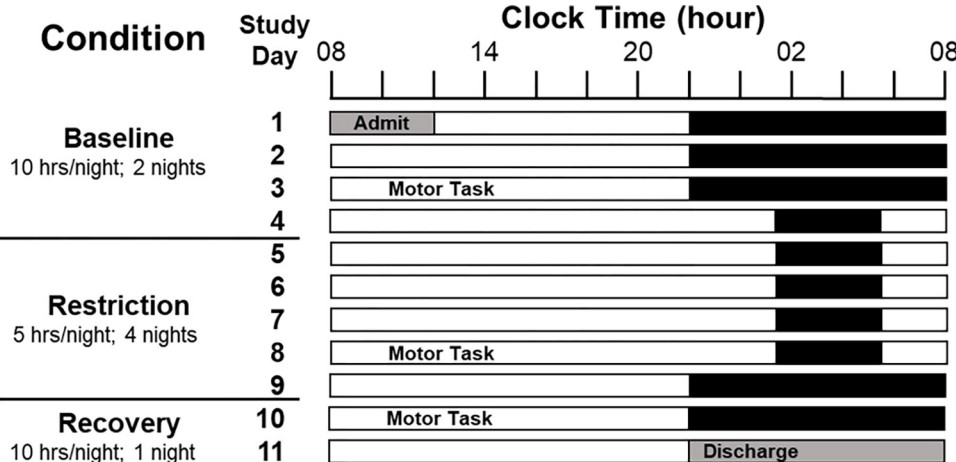

**Fig 1. Experimental timeline adapted with permission from Ness et. al., 2019.** White bars indicate time awake and black bars indicate time in bed. Participants completed motor task testing after 2 nights of 10-hours in bed (baseline condition); after four nights of 5-hours in bed (restriction condition); and for a third and final time after one night of 10-hours in bed (recovery condition).

condition, participants were restricted to five hours in bed for five nights during the sleep restriction condition. The final two nights of the study consisted of 10 hours in bed during recovery sleep. Hours spent sleeping was determined via polysomnography (PSG) and wrist actigraphy, however, the current analysis included only actigraphy data due to missingness of PSG for specific nights of interest in two participants. For the full duration of the study, food intake was strictly controlled according to a eucaloric diet consisting of weighed foods with predetermined macronutrient and micronutrient contents [26]. During the study, participants were permitted to engage in activities such as reading, completing puzzles, light stretching, and browsing the internet, provided no study procedures were scheduled at that time.

The motor force task took place once per study condition at 10:15 AM on days three, eight, and 10 of the inpatient stay. These corresponded to the second baseline day (day three of the study), the fourth sleep restriction day (day eight of the study), and the first recovery day (day 10 of the study). In the current analyses, baseline, sleep restriction, and recovery sleep refer to these three days, respectively.

## Measurement of sleep

Actigraphy data were downloaded with Philips Actiware software (versions 6.0.4. and 6.0.9.). At least two independent scorers (blinded to each other) determined "day" cut-point times, validity of days, and set sleep intervals using a previously validated procedure [29]. The scorers adjudicated each recording by verifying number of valid days, cut point, number of sleep intervals, and differences greater than 15 minutes in duration and wake after sleep onset for each sleep interval. The measure of interest calculated by actigraphy was total sleep time (TST) in hours for the sleep interval.

## Sleepiness scores

Self-reported sleepiness was assessed using the Karolinska Sleepiness Scale (KSS; [30] administered on a secure website (Research Electronic Data Capture, REDCap, versions 6.10.11 through 8.1.16; [31])). This is a 9-point scale with higher numbers indicating greater sleepiness: 1 = extremely alert; 3 = alert; 5 = neither alert nor sleepy; 7 = sleepy but no difficulty remaining awake; and 9 = extremely sleepy, fighting sleep. Participants may select even-number ratings, which have no descriptors, but represent sleepiness ratings in between the odd numbers. The KSS was administered 3–6 times on days three, eight, and 10 of the study, but for the current analysis, only the most proximal ratings to the motor force task were used (~1:30 PM).

## Motor force tasks

Precision grip strength of the dominant (right) hand was assessed by obtaining each participant's maximum voluntary contraction (MVC) using a pinch grip dynamometer (Lafayette Hydraulic Gauge, Lafayette, IN). The average of three 5-second trials determined each participant's MVC in newtons. Participants completed MVCs and the visually and memory-guided force control tasks on baseline, sleep restriction, and recovery sleep days.

Participants were seated in an upright chair (JedMed Straight Back Chair, St. Louis, MO) 127 cm from a 102 cm Samsung television that had a resolution of 1920 x 1080 and a refresh rate of 120 Hz. With their forearm resting at approximately 100 degrees of elbow flexion, participants used their thumb and index finger to form a pinch grip against two ELFF-B4 model load cells constructed from piezoresistive strain gauges (Measurement Specialties, Hampton, VA) (Fig 2A). Force data were collected by Coulbourn Instruments Type B V72-25B amplifiers at an excitation voltage of 5V. The voltage was transmitted via a 16-bit A/D converter and

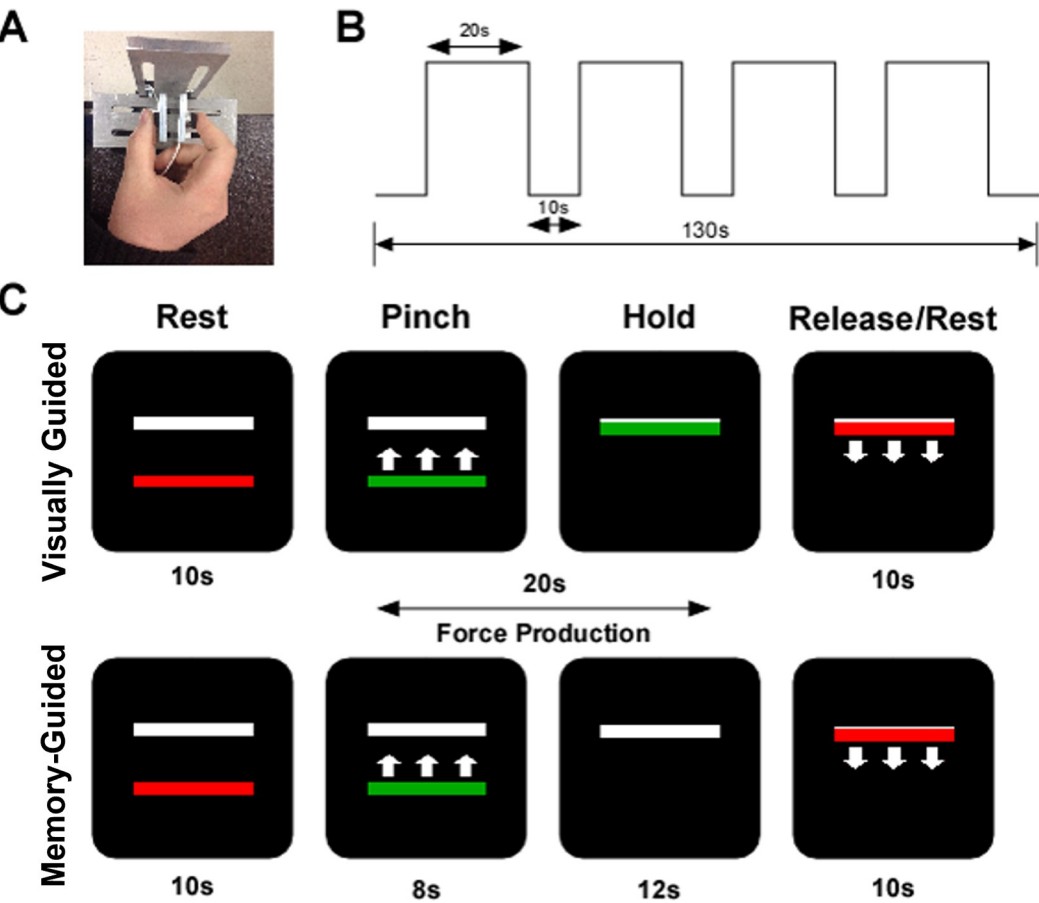

**Fig 2. Visually guided and memory-guided force paradigm.** A) The precision grip apparatus with load cells under the thumb and index finger; B) the experimental procedure was 130 s in length. Each block of 20 s of force was separated by 10 s of rest; C) the visual display contained two horizontal bars presented against a black background. The target bar (white) was stationary, and the red/green force bar provided real-time visual feedback. In the visually guided task, visual feedback was available for the duration of the trial. In the memory-guided task, the force bar disappeared for the last 12 s of the trial.

digitized at 62.5 Hz. The A/D board units were transformed to newtons using a calibration factor derived from known weights. The voltage range was -10 to 10V, and the A/D board was able to detect force levels as low as 0.0016 newtons. The summed output from the load cells was presented to the participant on the television screen in real time. Voltage data acquisition, voltage-to-force transformation, and stimuli presentation were all conducted using customized programs written in LabVIEW (National Instruments, Austin, TX). Stimuli were presented on the television screen.

The visually and memory-guided force tasks in the current study have been previously used to study visuomotor control in autism spectrum disorders [18, 32], attention-deficit/hyperactivity disorder [33], Parkinson's disease [34], and younger and older adults [17]. During the task, participants viewed two horizontal bars: a red/green force bar that moved up with increasing force and down with decreasing force, and a static white bar representing target force. The target white bar was set at 25% of the participant's MVC. The onset and offset of force production were cued by a color change of the moveable force bar. Green served as the go cue and red as the stop cue. Participants were instructed to produce force as quickly and as accurately as possible at the time of the color change from red to green and to keep the green bar at the target force level for the duration of the 20-second trial, until offset of force was

cued. As shown in Fig 2B, each run started and ended with 10 seconds of rest and included four 20-second trials of force with 10 seconds of rest in between each trial. During visually guided trials, the moveable force bar was visible for the duration of the trial, providing real-time visual feedback about performance. As shown in Fig 2C, during the memory-guided trials, the force bar disappeared for the last 12 seconds of the trial. Participants were instructed to continue producing force at the target level until the trial ended. Participants completed one run of four 20-second visually guided and one run of four 20-second memory-guided trials. The task order was counterbalanced across participants. All participants completed a brief practice session to become familiar with the timing and force output requirements of the task. The force time series data were digitally filtered using a fourth-order Butterworth filter with a 10 Hz low-pass cut-off frequency. We examined force during the last 12 seconds of each 20-second trial, which represents the time in which visual feedback was removed in the memory-guided condition. Force data were collected in newtons and were divided by the participant's MVC measured on the same day, multiplied by 100%. Therefore, the data were analyzed as a percent of MVC.

## Statistical analysis

All data were used in the analysis, but time in seconds and trial number were not included as factors. Therefore, the analysis design was repeated measures with two main design factors—day (baseline, sleep restriction, recovery sleep) and vision condition (visually guided and memory-guided)—and four potential covariates (race, age, TST, and KSS score).

We used a mixed effects multilevel approach to analyze the effects of sleep restriction and visual feedback on mean force produced in the last 12 seconds of each trial, normalized by MVC, in addition to modeling how the effects of day and vision condition varied across individuals. Linear mixed effects models employ a partial pooling method of data aggregation [35, 36], which allows all available data to be used (all time points and all trials per day per condition). The method shrinks the estimates toward the mean estimate, which includes but lessens the effect of the extreme outcomes in the data.

A series of models were estimated in a two-level multilevel framework using the lme4 package [36] in R [37] to model the mean force, normalized by MVC, across days and vision condition, nested within participants. All models were initially estimated using restricted maximum likelihood (REML). Models were compared using the Akaike's Information Criterion (AIC), where a lower AIC indicated a better fit to the data [38]. Within the best-fitting model, analyses of variance with Satterthwaite's method of determining degrees of freedom were used to determine if the interactions and main effects were significant [36, 39], where *a priori* significance for fixed effects was set at 0.05.

First, a series of random intercept models was estimated to understand the effects of the fixed design factors of the study on participants' mean force production during the last 12 s of each trial (S1 Table). These factors were day (baseline, sleep restriction, recovery sleep; baseline was the reference day) and vision condition (visually guided and memory-guided; visually guided was the reference condition). Second, a series of random intercept models with potential covariates was estimated (S2 Table). The covariates included race, age, TST, and KSS score, where age, TST, and KSS score were grand mean centered. Third, interactions were added to the resulting random intercept models with covariates that improved model fit by AIC (S3 Table). As this study was among the first to explore the effects of sleep restriction on continuous motor force production, we included interactions between terms to elucidate if and in what manner these covariates interacted with the factors of interest (day and vision condition). Fourth, we estimated a model allowing individual intercepts to vary across day and vision

condition to determine if sleep and vision conditions lead to difference in force variability (S4 Table). The best-fitting model of those estimated above, determined by lowest AIC (Lohse, 2020), would be deemed the final model, and the main effects and interactions of this final model would be evaluated.

## Results

### Participants

Fourteen healthy young adult males ($M \pm SD$, 22 ± 3 years, 9 white non-Hispanic, 3 Asian, 3 black non-Hispanic) participated in this study. Table 1 includes covariates and characteristics of the participants. Data were missing for KSS score for the baseline day for one participant and force data for the sleep restriction day for one participant.

### Grip force

The model including random slopes for the day and vision condition (AIC = 1.06x10^6) fit the data better than the model assuming no variation due to day and vision across individuals (AIC = 1.08x10^6). Therefore, random slopes across repeated measures factors were included in the final model.

   The best-fitting model, shown in Table 2, included main effects for day, vision condition, KSS score, and the three-way interaction among day, vision condition, and KSS score. The final model also included random slopes for both day and vision condition. Fig 3A shows the mean force produced during the final 12 s for each participant and averaged across participants. Although time was not a factor in the model, Fig 3B displays force production (averaged across participants), by day and vision condition, over the final 12 s (which was the time period analyzed from each trial). Table 3 includes the estimated marginal means of force output by vision condition and by day.

   When the model indicated significant interactions or main effects, Satterthwaite pairwise comparisons were conducted. As shown in Fig 3A, there was an effect of vision condition such

**Table 1. Participant characteristics.**

| Participant | KSS Baseline | KSS Restriction | KSS Recovery | MVC Baseline (N) | MVC Restriction (N) | MVC Recovery (N) |
|---|---|---|---|---|---|---|
| Participant 1 | 4 | 9 | 5 | 20.02 | 22.24 | 26.68 |
| Participant 2 | 8 | 9 | 7 | 48.90 | 44.48 | 73.40 |
| Participant 3 | 5 | 7 | 5 | 46.71 | 26.69 | 33.36 |
| Participant 4 | 1 | 3 | 1 | 66.72 | 66.72 | 66.72 |
| Participant 5 | N/A | 5 | 3 | 73.40 | 75.62 | 77.84 |
| Participant 6 | 3 | 5 | 3 | 73.40 | 51.15 | 71.17 |
| Participant 7 | 1 | 8 | 1 | 40.03 | 26.68 | 37.81 |
| Participant 8 | 7 | 9 | 7 | 26.69 | 22.24 | 26.69 |
| Participant 9 | 4 | 5 | 5 | 57.83 | 46.71 | 37.80 |
| Participant 10 | 3 | 4 | 3 | 46.71 | 68.95 | 60.05 |
| Participant 11 | 6 | 6 | 3 | 66.72 | 57.83 | 60.05 |
| Participant 12 | 4 | 9 | 4 | 53.38 | 53.38 | 64.50 |
| Participant 13 | 2 | 4 | 4 | 42.26 | 48.93 | 48.93 |
| Participant 14 | 1 | 5 | 3 | 93.41 | 82.29 | 104.53 |
| **Mean** | **4** | **6** | **4** | **54.01** | **49.57** | **56.40** |
| **Standard Deviation** | **2** | **2** | **2** | **19.63** | **19.77** | **22.41** |

Note. KSS, Karolinska Sleepiness Scale; MVC, maximum voluntary contraction; N, newtons.

**Table 2. Results of the multilevel mixed-effects model estimating mean force (%MVC).**

| Variable | Estimate (SE) |
|---|---|
| Intercept (Day = Baseline, Vision = VG, KSS = 4) | 24.80 (0.16)*** |
| Day (Restriction vs. Baseline) | -0.84 (0.38)* |
| Day (Recovery vs. Baseline) | 0.15 (0.23) |
| Vision (MG vs. VG) | -0.71 (0.29)* |
| KSS | -0.02 (0.06) |
| Day (Restriction vs. Baseline) x Vision (MG vs. VG) | 0.10 (0.03)** |
| Day (Recovery vs. Baseline) x Vision (MG vs. VG) | 0.03 (0.02) |
| Day (Restriction vs. Baseline) x KSS | 0.12 (0.12) |
| Day (Recovery vs. Baseline) x KSS | 0.02 (0.11) |
| Vision (MG vs. VG) x KSS | -0.21 (0.01)*** |
| Day (Restriction vs. Baseline) x Vision (MG vs. VG) x KSS | 0.31 (0.01)*** |
| Day (Recovery vs. Baseline) x Vision (MG vs. VG) x KSS | 0.31 (0.01)*** |
| AIC | 1059064.60 |
| BIC | 1059303.54 |
| Log-likelihood | -529509.30 |
| Num. obs. | 239985 |
| Num. groups: Participant | 14 |
| Var: Participant (intercept) | 0.33 |
| Var: Participant Day (Restriction) | 1.03 |
| Var: Participant Day (Recovery) | 0.73 |
| Var: Participant Vision (Memory-Guided) | 1.14 |
| Cov: Participant (Intercept) Day (Restriction) | -0.03 |
| Cov: Participant (Intercept) Day (Recovery) | -0.42 |
| Cov: Participant (Intercept) Vision (Memory-Guided) | -0.08 |
| Cov: Participant Day (Restriction) x Day (Recovery) | 0.23 |
| Cov: Participant Day (Restriction) x Vision (Memory-Guided) | -0.58 |
| Cov: Participant Day (Recovery) Vision (Memory-Guided) | 0.01 |
| Var: Residual | 4.82 |

Note. SE = standard error; VG = Visually Guided; MG = Memory-Guided; KSS = Karolinska Sleepiness Scale.

***$p < 0.001$

** $p < 0.01$

*$p < 0.05$.

that participants produced less force in the memory-guided condition than in the visually guided condition at baseline ($B = -0.711\%$, $p = 0.027$). There was an effect of day within the visually guided condition such that when visual feedback was present, participants produced less force during the sleep restriction day than during the baseline day ($B = -0.838\%$, $p = 0.041$). There was an interaction between day and vision condition such that participants did not produce different force during the sleep restriction day than during the baseline day during memory-guided force production ($B = 0.96\%$, $p = 0.005$) (Fig 3A). There was no effect of recovery sleep on visually guided force production ($B = 0.15\%$, $p = 0.520$) or on memory-guided production ($B = 0.03\%$, $p = 0.224$) (Fig 3A).

There were significant three-way interactions between day, vision condition, and KSS score, shown in Fig 4. Follow-up general linear models were run to estimate the effect of KSS score in each vision condition on each day. The relationship between KSS score and visually guided force production did not differ from baseline to sleep restriction to recovery sleep.

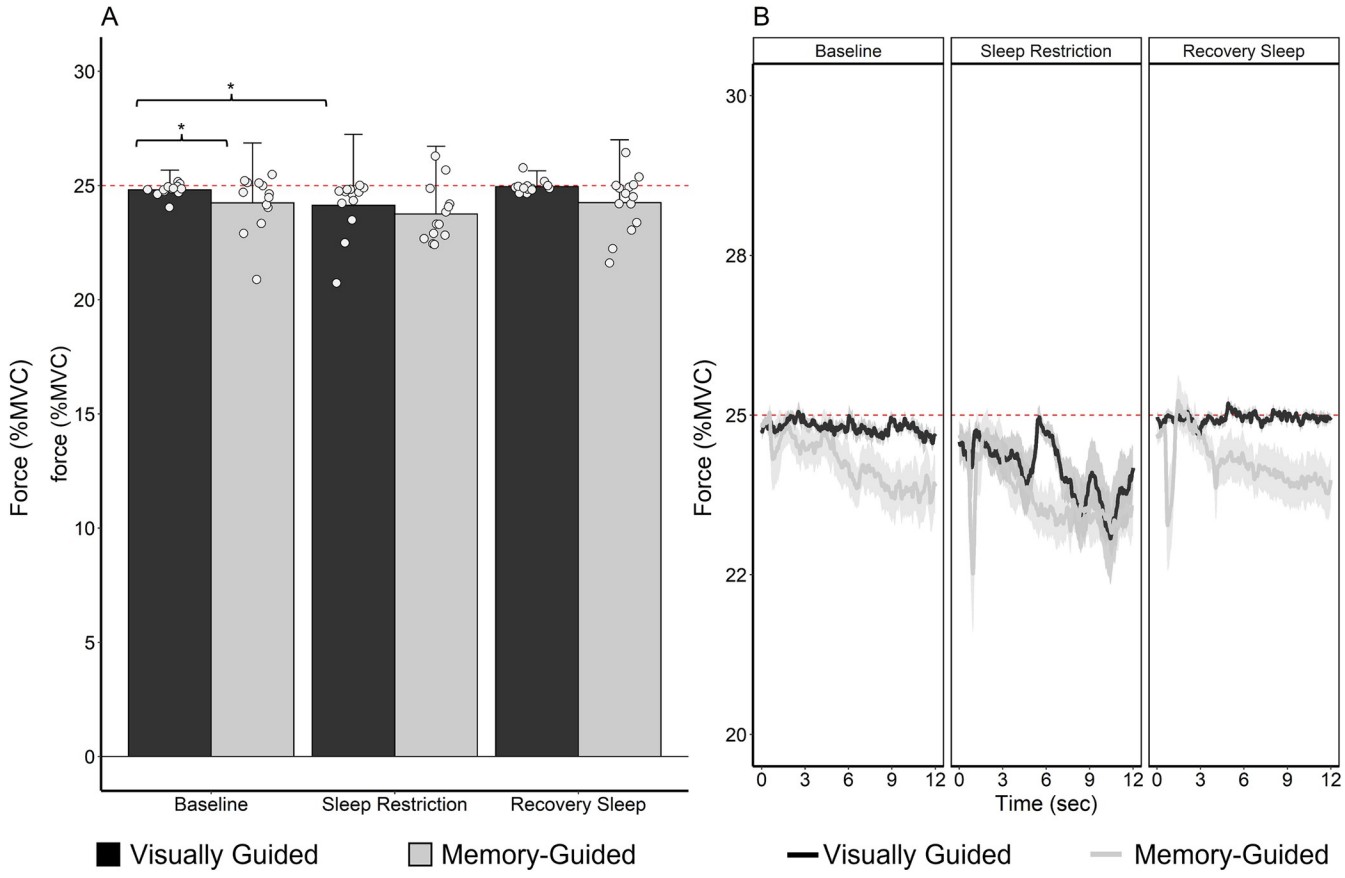

**Fig 3. Force production as a percentage of maximum voluntary contraction (MVC) plotted by day and vision condition.** The red dashed line indicates the target force (25% MVC). A) Average force production over the last 12 s of each trial. The black bar indicates the visually guided condition, and the grey bar indicates the memory-guided condition. Error bars indicate the standard deviation of the mean. Open circles represent each individual participant's mean force. ***$p < 0.001$; ** $p < 0.01$; *$p < 0.05$. B) Force production over the last 12 s of each trial. The dark black line indicates the visually guided condition, and the grey line indicates the memory-guided condition. Shaded areas represent the standard error of the mean.

However, the relationship between KSS score and memory-guided force production significantly changed after sleep restriction ($p = 0.002$) and after recovery sleep ($p = 0.031$) relative to baseline. After sleep restriction and recovery sleep, higher KSS scores (i.e., greater self-reported

**Table 3. Estimated marginal means and estimated marginal trends for force (% MVC).**

| Estimated Marginal Means | | | |
|---|---|---|---|
| | Baseline | Sleep Restriction | Recovery Sleep |
| Visually Guided | 24.8% (0.2%) | 24.0% (0.4%)* | 24.9% (0.1%) |
| Memory-guided | 24.0% (0.3%)† | 23.5% (0.3%) | 24.3% (0.3%) |
| **Estimated Marginal Trends By KSS Score** | | | |
| | Baseline | Sleep Restriction | Recovery Sleep |
| Visually Guided | -0.02 (0.06) | 0.10 (0.12) | -0.003 (0.07) |
| Memory-guided | -0.23 (0.06) | 0.21 (0.12) * | 0.09 (0.07) * |

Note. Force production was measured in percent of maximum voluntary contraction in newtons. Values in paratheses are standard error of the mean. Asterisks indicate significant difference between a given day and Baseline, and daggers indicate significant differences between Visually and memory-guided conditions.

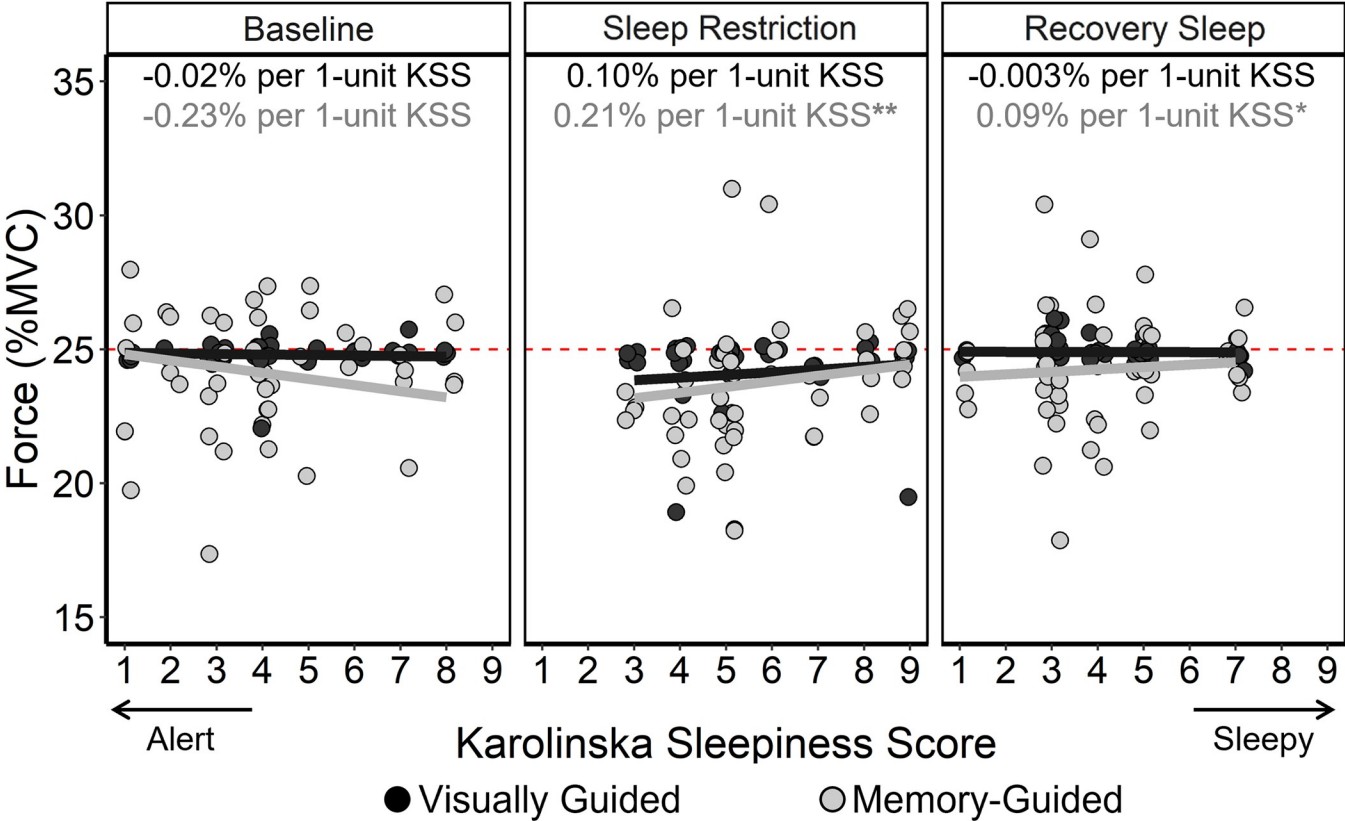

**Fig 4. Force production by day and vision condition as a percentage of maximum voluntary contraction (MVC), as related to Karolinska Sleepiness Scale (KSS) score.** The red dashed line indicates the target force (25% MVC). The black circles and lines indicate the visually guided condition, and the grey points and lines indicate the memory-guided condition. Each circle indicates a participants' average force during each trial (4 trials per vision condition). Lines indicate the relationship between KSS score and force relative to MVC. Asterisks indicate difference in slopes between baseline and the respective day. ** $p < 0.01$; * $p < 0.05$.

sleepiness) were associated with higher force production during the memory-guided condition relative to baseline.

To describe the heterogeneity of variance across levels of the repeated measures factors, we investigated the model residuals across levels of day and vision condition, as shown in Fig 5 and described in Table 4. Visual inspection showed that there was larger variability in the residuals (as measured through standard deviation) on the sleep restriction day than at baseline and after recovery sleep, and there was larger variability in residuals during memory-guided than during visually guided force production. However, the effect of sleep restriction on model residual variability was considerably larger during visually guided then during memory-guided force production. This effect was only observed when one participant (Participant 12) was removed from the model. Participant 12 was the only participant with considerable variability in the visually guided condition on the sleep restriction day (standard deviation of 6.7% MVC compared to the sample standard deviation (without Participant 12) of 2.4% MVC). The bottom section of Table 4 includes the variability by day and vision condition with and without Participant 12 included.

Finally, to further evaluate model fit, we examined the model residual variability described in Table 2 (Var: Residual). Specifically, after accounting for the between-person fixed and random effects, the standard deviation of the residuals was 2.19% MVC, demonstrating substantial remaining variance (relative to the fixed effects coefficients) in force production after accounting for day, vision condition, KSS, and MVC.

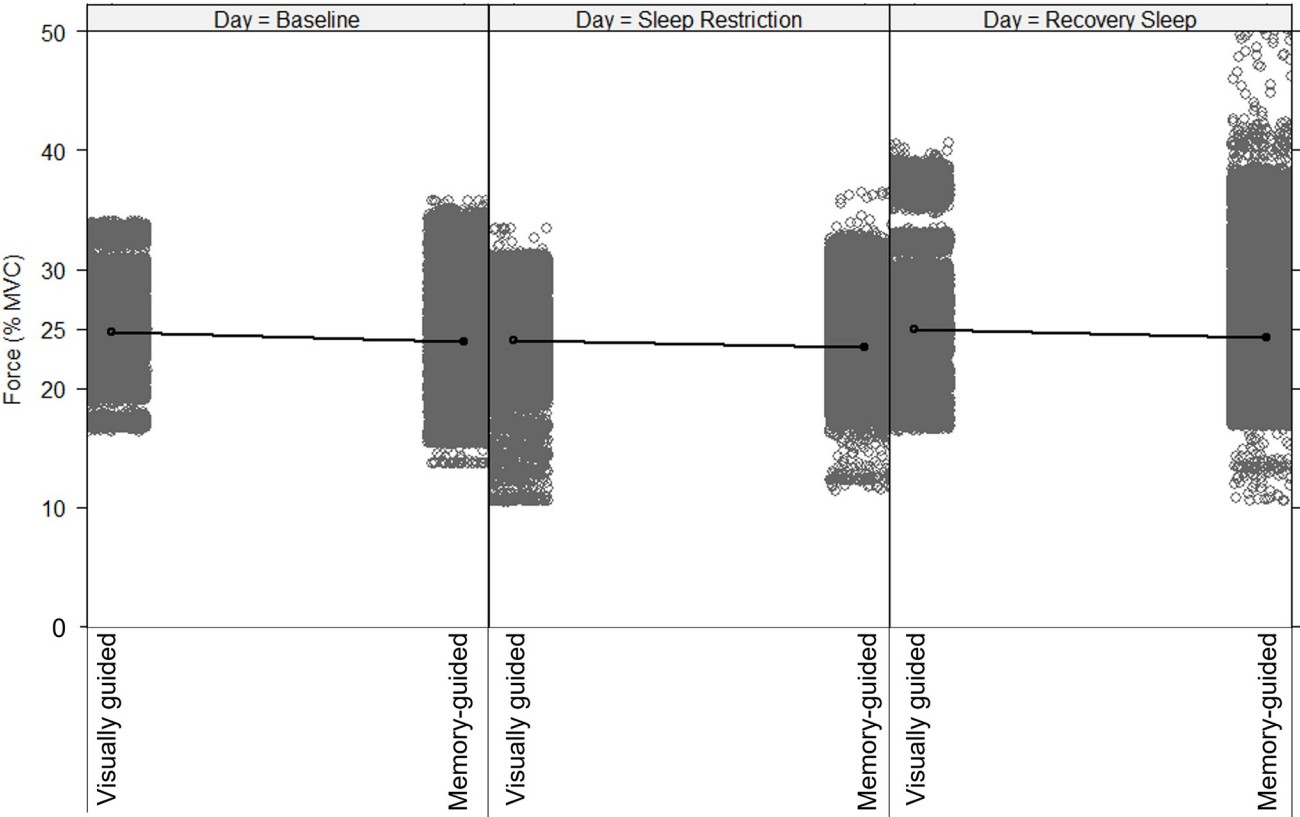

**Fig 5. Residuals of the final model across day and vision condition, demonstrating that residual variability in force production was higher in the memory-guided than the visually guided condition after sleep restriction and recovery sleep across participants.**

## Discussion

The purpose of this study was to investigate the effects of sleep restriction and visual feedback on force production in healthy young adult males. This study had two main findings: (1)

**Table 4. Standard deviation of the residuals of force production (% MVC) for experimental conditions, with and without Participant 12.**

| All Participants (N = 14) | | |
|---|---|---|
|  | **Visually Guided** | **Memory-Guided** |
| **Baseline** | 0.93% | 2.37% |
| **Sleep Restriction** | 2.90% | 2.75% |
| **Recovery** | 0.74% | 2.48% |
| N = 13 | | |
| **Baseline** | 0.80% | 2.37% |
| **Sleep Restriction** | 2.32% | 2.84% |
| **Recovery** | 0.75% | 2.45% |

Note. One participant of the fourteen had considerable force variability in the visually guided condition during the sleep restriction day compared to other participants. When that participant was removed from the model (bottom portion of the table), there was a noticeably higher residual variability in the memory-guided condition compared to the visually guided condition during sleep restriction, paralleling the results of the baseline and recovery days.

Visually guided force production was more sensitive to sleep loss than memory-guided force production, and (2) During memory-guided force production, higher self-reported sleepiness was related to higher force after sleep restriction and recovery sleep.

Visually guided force production was more sensitive to sleep loss than memory-guided force production. Specifically, in the visually guided condition, participants produced less force after four nights of sleep restriction compared to baseline. In contrast, in the memory guided condition, participants' force output did not differ across baseline, sleep restriction, and sleep recovery. The finding that visually guided force production is more sensitive (to sleep restriction) than memory-guided force production is consistent with literature reporting that memory is less affected by sleep loss than simple attention [6]. Indeed, the maintenance of visually guided force output at a specified target level requires sustained attention and feed-back-based corrections [40], both of which may be sensitive to sleep loss. Performing a continuous visually guided force task may be more similar to responding during real-world tasks, such as operating a motor vehicle, compared to a discrete button-pressing tasks commonly employed in sleep loss paradigms.

The effect of sleep restriction on force production varies across people. Figs 3 and 4, and Table 4, visualize and report the between-participant variability in the amount of force produced. Between-participant variability in the memory-guided force condition was consistent across days (baseline, sleep restriction, and recovery sleep). However, the between-participant variability in the visually guided condition was nearly three times higher after sleep restriction than at baseline and after recovery sleep. Considering that visually guided force control requires feedback-based motor corrections [40], the results suggest that for some individuals, feedback-based force control may be more sensitive to sleep restriction than memory-guided force control. The current study's findings of interindividual effects of sleep restriction on performance corroborate previous research using other cognitive tasks [7]. Future studies could explore the effect of sleep restriction as a longitudinal (over the 12 seconds of gripping) factor in multilevel models rather than as a categorical factor, to determine the potentially compounding effect of sleep loss on visually and memory-guided motor output across time.

Although visually guided force production was impaired after sleep restriction, this was not explained by self-reported sleepiness. Self-reported sleepiness did not relate to force produced during the visually guided condition at baseline, after sleep restriction, or after recovery sleep. As shown in Fig 3, sleep restriction reduced visually guided force output in some individuals more than others; however, self-reported sleepiness was not associated with force output. This finding suggests that the KSS may not be a useful measure to predict visually guided motor output. Self-reported sleepiness did, however, predict the amount of force produced in the memory-guided condition after sleep restriction and after recovery sleep. Specifically, after sleep restriction and recovery sleep, sleepiness predicted more force in the memory-guided condition (Fig 4). Sleepier, sleep-restricted young adults may be less reliant on visual feedback and more reliant on proprioceptive feedback during memory-guided force production. Considering that visual tracking and attention are impaired after sleep restriction [6, 41], it is possible that sleepier sleep-restricted young adults may trigger a compensatory feedforward strategy while completing the memory-guided force production task, which is more cognitively taxing than the visually guided task [6, 41]. Higher sleepiness could induce a compensatory strategy to mitigate a larger perceived decrement in their force production when visual feedback is removed due to reliance on proprioceptive feedback [7]. While self-reported sleepiness predicted force output, TST, measured by actigraphy, did not. As expected, the pseudo-$R^2$ for day predicting TST was 0.93, and because TST and day were so highly correlated, TST did not improve force estimation beyond that predicted by day.

## Limitations

This study had some limitations. First, the force task asked participants to maintain a grip force output of 25% of MVC. Grip force of 20–40% MVC is important for activities of daily living such as writing, unlocking a door, and tying shoes [42], but it may not be directly applicable to other activities requiring higher or lower grip force (such as operating heavy machinery). Future studies should investigate the effect of sleep restriction on various motor force tasks to better understand the wholistic effect of sleep restriction on movement. Also, we modeled force data without including a longitudinal effect (over the 12 s of gripping). As this was the first study exploring the effect of sleep restriction on grip force output, we were interested in the overall effect (or lack thereof) of sleep restriction and recovery sleep on motor output. Future studies may model continuous force production over time to determine if the effects of sleep restriction we found here change with duration of force production.

## Conclusion

Sleep restriction impairs visually but not memory-guided continuous force output. Both mean force and between-participant variability in force are more sensitive to sleep restriction when visual feedback is present. Also, after sleep restriction and recovery sleep, higher levels of self-reported sleepiness are associated with higher force production in the memory-guided condition. Taken together, these findings demonstrate that sleep restriction impacts visually guided force output; however, the response to sleep restriction is heterogeneous, in spite of having tested a homogeneous sample of young, healthy males, under well-controlled experimental conditions. Additionally, sleep restriction impairs visually guided force output regardless of perceived sleepiness; however, in the memory-guided condition, force becomes more impaired with increasing levels of perceived alertness.

## Supporting information

**S1 Table. Random intercept models.** Note. Values given in Estimate (Standard Error); VG = Visually Guided; MG = Memory-Guided. First series of random intercept models estimated to understand the effects of the fixed design factors of the study on percent of a participant's MVC during the last 12 sec of each trial. The best-fitting model of the series—determined by AIC—was Model 4: Day*Vision interactions. ***$p < 0.001$; ** $p < 0.01$; *$p < 0.05$.
(DOC)

**S2 Table. Random intercept models with potential covariates.** Note. Values given in Estimate (Standard Error); VG = Visually Guided; MG = Memory-Guided; TST = Total Sleep Time; KSS = Karolinska Sleepiness Scale. Second series of random intercept models with potential covariates. The best-fitting model of the series—determined by AIC—was Model 4: KSS. ***$p < 0.001$; ** $p < 0.01$; *$p < 0.05$.
(DOCX)

**S3 Table. Random intercept models with study design and covariates.** Note. Values given in Estimate (Standard Error); VG = Visually Guided; MG = Memory-Guided; KSS = Karolinska Sleepiness Scale. Third series of random intercept models with interactions between study design variables and covariates that improved model fit by AIC. The best-fitting model of the series—determined by AIC—was Model 2: Main effects, all interactions. ***$p < 0.001$; ** $p < 0.01$; *$p < 0.05$.
(DOCX)

**S4 Table. Random slopes models with study design and KSS.** Note. Values given in Estimate (Standard Error); VG = Visually Guided; MG = Memory-Guided; KSS = Karolinska Sleepiness Scale. Fourth series of models, comparing random slopes for day and vision and no random slopes. The best-fitting model of the series—determined by AIC—was Model 4: Random Slopes on Day and Vision. ***p < 0.001; ** p < 0.01; *p < 0.05.
(DOCX)

**S1 Data. Demographics, day, total sleep time, KSS, MVC, vision condition, trial, time in seconds, and force in newtons.**
(CSV)

## Acknowledgments

We thank the individuals who participated in the study. We also thank the students and staff of the Chang and Buxton laboratories for their assistance with study procedures, particularly Nicole Nahmod for participant recruitment; the staff of the Clinical Research Center; Chloe House, PhD, who conducted the psychological interviews; and study collaborators.

## Author Contributions

**Conceptualization:** Anne-Marie Chang, Kristina A. Neely.

**Data curation:** Sarah A. Brinkerhoff, Kristina A. Neely.

**Formal analysis:** William M. Murrah.

**Funding acquisition:** Anne-Marie Chang.

**Methodology:** William M. Murrah, Kristina A. Neely.

**Visualization:** Sarah A. Brinkerhoff, William M. Murrah.

**Writing – original draft:** Sarah A. Brinkerhoff, Kristina A. Neely.

**Writing – review & editing:** Sarah A. Brinkerhoff, Gina M. Mathew, William M. Murrah, Anne-Marie Chang, Jaimie A. Roper, Kristina A. Neely.

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
