## [Decision Letter · Decision Letter 0]

2 May 2022

PONE-D-22-02851Sleep Restriction Impairs Visually and Memory-Guided Force ControlPLOS ONE

Dear Dr. Brinkerhoff,

Thank you for submitting your manuscript to PLOS ONE. After careful consideration, we feel that it has merit but does not fully meet PLOS ONE’s publication criteria as it currently stands. Therefore, we invite you to submit a revised version of the manuscript that addresses the points raised during the review process.

We look forward to receiving your revised manuscript.

Kind regards,

Kenichi Shibuya, Ph.D.

Academic Editor

PLOS ONE

Journal Requirements:

Additional Editor Comments (if provided):

I have completed my evaluation of your manuscript. The reviewers recommend reconsideration of your manuscript following major revision. I invite you to resubmit your manuscript after addressing the comments below. When revising your manuscript, please consider all issues mentioned in the reviewers' comments carefully: please outline every change made in response to their comments and provide suitable rebuttals for any comments not addressed. Please note that your revised submission may need to be re-reviewed.

Reviewers' comments:

Reviewer's Responses to Questions

**Comments to the Author**

1. Is the manuscript technically sound, and do the data support the conclusions?

Reviewer #1: Partly

Reviewer #2: No

2. Has the statistical analysis been performed appropriately and rigorously? 

Reviewer #1: I Don't Know

Reviewer #2: I Don't Know

3. Have the authors made all data underlying the findings in their manuscript fully available?

Reviewer #1: Yes

Reviewer #2: Yes

4. Is the manuscript presented in an intelligible fashion and written in standard English?

Reviewer #1: Yes

Reviewer #2: Yes

5. Review Comments to the Author

Reviewer #1: I reviewed the manuscript “Sleep restriction impairs visually and memory-guided force control” by Brinkerhoff and Colleagues. The authors report results from a study in which male participants produced isometric force between the index finger and thumb at various times as they underwent a sleep restriction paradigm. These participants’ ability to produce a steady level of force, and to maintain it under visual feedback, decreased when they were sleep-restricted. Overall I think the experiment described is technically sound; my main concern is that the unconventional statistical methods make the results difficult to understand.

Main:

1. The treatment of time-series data is confusing, although I think it is alluded to around Line 191. Does this mean that the input data for the model fitting are all timepoints for each trial (e.g. 62.5x12 = 750 samples)? This seems to be the case given the number of observations in Table 2, but I am concerned that each datapoint is treated as independent to be fit by the model, when there is time series dependence between certain sets of data points, e.g. if the participant is producing 24% MVC at time t, they are more likely to produce 24.3% than 5% at t+1. I think the authors should more clearly lay out how time-series data are treated and explain why it is alright not to include a “time in trial” or similar covariate in the model. Overall I would strongly suggest the authors use more summarized data rather than the current approach.

2. The statistics are also confusing because the statistical analysis is not carried out on the actual interpreted outcome variables. For example, the variability of participants’ force data is referred to in terms of magnitude of standard deviation changing in the results, but the actual tests are on the size of model residuals. While these are related, it seems like testing the standard deviations themselves would be clearer, especially because it’s unclear how standard deviation is calculated in light of the handling of time series data: what is “N” the denominator of SD? SD across participants and trials?

2a. I think the visual appearance that more sleepiness results in “better” force production -- that is: closer to the target level (Figure 4B) -- is another counterintuitive outcome of this analysis design. Similarly force production is not averaged but the data in Fig 3A still are means and error bars are calculated based on trial-by-trial or participant-by-participant SEM, although these measures are not what the statistics tested exactly.

3. The justification for use of a “mixed effects approach” (line 186) is confusing because it makes it seem like there is a direct association between variability of motor output and model variance, although the association is not so direct. Further, the approach that ends up being used here still has the same requirements, and homoscedasticity is never assessed. Overall, I think that more conventional statistics (perhaps using data transformation if necessary) would make the results much easier to interpret.

Minor:

1. My impression is that the association between working memory and memory-guided force production is usually related to how much force production changes after visual feedback is removed (e.g. Vaillancourt & Russell 2002, EBR). Is there a reason this measure was not assessed? This seems potentially important if some of the significance of the study is related to working memory.

Reviewer #2: The authors showed that sleep restriction negatively affected accuracy and variability of force production. One night of recovery sleep was effective to return to the baseline performance only for the visual-guided condition. Although the study results were interesting, this reviewer has concerns, including the details of statistical analysis and how those were reported, and the study rationale.

Specific comments:

Introduction

- # 13: The author mentioned that sleep restriction negatively affects working memory. However, there was a lack of explanations about what working memory is and why investigating the effects of sleep restriction on working memory is essential. Please provide more explanation and supporting evidence that supports your study rationale.

- # 17: The participants in this study were young, but the major reference that explains the study rationale comes from a study that examined older adults.

- # 25 – 31: Many daily living tasks especially involving manual dexterity (e.g., hands) require continuous modulation of force output, but the experimental task used in this study was not the case. Please justify how the experimental task used in this study represents daily motor tasks.

Methods

- #207: The author stated that analyses of variance with Satterthwaite’s method was used. More detail is needed to explain the aim of using the method, which could benefit readers who are not familiar with the method.

- #215: It seemed that there were three main factors, but the statistical section would be improved by having more details about what final models the author used and how the results will be interpreted in the Results section.

Results

- Readers could better understand when more direct explanations for the statistical results were included. The authors utilized a mixed model to examine the main and interaction effects with coefficient values in Table 2. However, it would be challenging to interpret the entire table and then match the results in the table with what the authors described their findings in the results section (also for the Discussion). For example, the authors may first describe the best model and the results used to describe the mean differences with statistical results (e.g. p values) in the Results (e.g., Figure 3).

- #250: Please provide numbers (“these effects were fairly small in magnitude”).

- Table 2: It was challenging to understand this table. As mentioned for the statistical section, there were three main factors (please correct me if I am wrong), and I expected some information about significant interaction between Day (three levels; Baseline, Restriction, and Recovery) and Vision (two levels; Visually-Guided and Memory-Guided). The author should provide more details about how Table 2 can be interpreted..

- When an interaction effect is significant, pair-wise comparisons are usually the next step, as shown in previous studies using ANOVA, but those statistical results with p values were not included. Did not the mixed model provide that information? A similar question was raised in Figure 3 also. Statistical results from within-group comparisons only seemed to be shown, so please provide statistical results for the other paired comparisons (e.g., Visually vs. Memory in the baseline).

- # 285: The authors interpreted the main effect of vision condition to describe their finding, but there was no further information on it. Adding a supporting statistical result can strengthen the sentence.

- Figure 5: Most results were reported by force relative to MVC (% MVC), but in Figure 5 the data were in newtons. Please clarify what values (e.g., N or % MVC) were used in that Figure. Could the normalization affect the findings?

Discussion

- The study findings were from an isometric pinching force production at a specific constant force output (25% MVC). The authors needed to address how the findings can be generalizable in other force production tasks that include different force production types (e.g., concentric), different amounts of force levels (less or higher than 25% MVC), and force production trajectories (e.g., continuous control of force level). Adding that information in a separate study limitation section can address this issue.

6. PLOS authors have the option to publish the peer review history of their article (what does this mean?). If published, this will include your full peer review and any attached files.

Reviewer #1: No

Reviewer #2: No

---

## [Author Response · Author response to Decision Letter 0]

24 Jun 2022

Editor Comments (if provided):

I have completed my evaluation of your manuscript. The reviewers recommend reconsideration of your manuscript following major revision. I invite you to resubmit your manuscript after addressing the comments below. When revising your manuscript, please consider all issues mentioned in the reviewers' comments carefully: please outline every change made in response to their comments and provide suitable rebuttals for any comments not addressed. Please note that your revised submission may need to be re-reviewed.

REVIEWER #1: 

I reviewed the manuscript “Sleep restriction impairs visually and memory-guided force control” by Brinkerhoff and Colleagues. The authors report results from a study in which male participants produced isometric force between the index finger and thumb at various times as they underwent a sleep restriction paradigm. These participants’ ability to produce a steady level of force, and to maintain it under visual feedback, decreased when they were sleep-restricted. Overall I think the experiment described is technically sound; my main concern is that the unconventional statistical methods make the results difficult to understand.

Author Response:

We thank the reviewer for their helpful comments and suggestions to enhance our manuscript. We have addressed each individual comment below. However, before doing so, we would like to mention two major revisions in response to reviewer feedback. First, we normalized force output to each participant’s maximum voluntary contraction (MVC) before entering force output into the statistical model. As a result, MVC has been removed from the model. We believe this change helps to clarify the Results. This change in our analysis affected the Results, such that the relationship between sleepiness and visual condition depended on the day; higher self-reported sleepiness after sleep restriction and recovery sleep led to smaller memory-guided force.

Main comments:

1. The treatment of time-series data is confusing, although I think it is alluded to around Line 191. Does this mean that the input data for the model fitting are all timepoints for each trial (e.g. 62.5x12 = 750 samples)? This seems to be the case given the number of observations in Table 2, but I am concerned that each datapoint is treated as independent to be fit by the model, when there is time series dependence between certain sets of data points, e.g. if the participant is producing 24% MVC at time t, they are more likely to produce 24.3% than 5% at t+1. I think the authors should more clearly lay out how time-series data are treated and explain why it is alright not to include a “time in trial” or similar covariate in the model. Overall, I would strongly suggest the authors use more summarized data rather than the current approach.

Author Response:

We appreciate the reviewer indicating the importance of considering time within our models. Yes, we used all data points within the models, and then estimated the effects of two design factors: day and vision condition. In this case, as the reviewer suggests, each data point was treated as independent from the point(s) before and after it in time. We agree that the time-series nature of the force data means that data samples close in time to one another are correlated with one other. One option to handle the data, as the reviewer suggests, is to average the data over the 12 seconds and across trials, and each participant would have a single data point per day, per vision condition (complete pooling). However, this method of pooling the data gives equal weight to extreme values in the outcome. The alternative method, used in our paper, is to allow the mixed models to aggregate the data across time, but not including time as a factor in the models. Accordingly, the linear mixed models also use averages, but they calculate those averages in the modeling process using partial pooling. This approach has three advantages: It 1) uses all available information in the data (all time points and trials within a day and condition), 2) weights the averages by the amount of information available (this is not a big benefit in this study as people have almost exactly the same number of time points in each trial), and 3) shrinks the estimates toward the mean estimate, which still includes, but lessens the effect of, extreme outcomes observed in the data (Bates, 2015; Gelman & Hill, 2008). Partial pooling assumes that each datapoint is some deviation from the average (i.e., regression to the mean) and applies less weight to datapoints that are not removeable outliers by design and should therefore be included in the dataset, which would otherwise pull the average in a direction that is not indicative of the participants’ true average force produced (Gelman & Hill, 2008). Therefore, the mixed models approach allows us to use all datapoints while more accurately summarizing the participant’s true force output (per day, per vision condition) than we would get if we manually averaged across design factors. While complete pooling and partial pooling both offer ways to summarize the data across time and trials, we believe partial pooling is more ecologically valid in this case.

This benefit is now emphasized in the statistical methods section of the manuscript:

Line 183-191: “All data were used in the analysis, but time in seconds and trial number were not included as factors. Therefore, the analysis design was repeated measures with two main design factors: day and vision condition. We used a mixed effects multilevel approach to analyze the mean effects of sleep restriction and visual feedback on motor force production, in addition to modeling how the effects of day and vision condition varied across individuals. Linear mixed effects models employ a partial pooling method of data aggregation (Gelman, 2006; Bates, 2015), which allows all available data to be used (all time points and all trials per day per condition). The method shrinks the estimates toward the mean estimate, which includes but lessens the effect of the extreme outcomes in the data.”

Additionally, we did not include a variable for time-in-trial because the question posed by this study was not longitudinal (over the 12 seconds of gripping) in nature; rather, we were interested in the overall effect of sleep restriction and recovery sleep on continuous force production.

2. The statistics are also confusing because the statistical analysis is not carried out on the actual interpreted outcome variables. For example, the variability of participants’ force data is referred to in terms of magnitude of standard deviation changing in the results, but the actual tests are on the size of model residuals. While these are related, it seems like testing the standard deviations themselves would be clearer, especially because it’s unclear how standard deviation is calculated in light of the handling of time series data: what is “N” the denominator of SD? SD across participants and trials?

Author Response:

We have revised the manuscript wording to make clearer the outcomes we examined. We examined the variation across participants estimated by the residuals, not the variation within participants.

The Results section now reads:

Line 308-321: “To describe the heterogeneity of variance across levels of the repeated measures factors, we investigated the model residuals across levels of day and vision condition, as shown in Figure 5 and described in Table 4. Visual inspection showed that there was larger variability in the residuals (as measured through standard deviation) on the sleep restriction day than at baseline and after recovery sleep, and there was larger variability in residuals during memory-guided than during visually guided force production. However, the effect of sleep restriction on model residual variability was considerably larger during visually guided then during memory-guided force production. This effect was only observed when one participant (Participant 12) was removed from the model. Participant 12 was the only participant with considerable variability in the visually guided condition on the sleep restriction day (standard deviation of 6.7% MVC compared to the sample standard deviation (without Participant 12) of 2.4% MVC). The bottom section of Table 4 includes the variability by day and vision condition with and without Participant 12 included.”

Line 336-341: “Finally, to further evaluate model fit, we examined the model residual variability described in Table 2 (Var: Residual). Specifically, after accounting for the between-person fixed and random effects, the standard deviation of the residuals was 2.19% MVC, demonstrating substantial remaining variance (relative to the fixed effects coefficients) in force production after accounting for day, vision condition, KSS, and MVC.”

2a. I think the visual appearance that more sleepiness results in “better” force production -- that is: closer to the target level (Figure 4B) -- is another counterintuitive outcome of this analysis design. Similarly force production is not averaged but the data in Fig 3A still are means and error bars are calculated based on trial-by-trial or participant-by-participant SEM, although these measures are not what the statistics tested exactly.

Author Response:

We agree that this finding is interesting and initially counterintuitive. The revised Discussion now includes more emphasis on this finding:

Line 386-396: “Specifically, after sleep restriction and recovery sleep, sleepiness predicted more force in the memory-guided condition (Figure 4). Sleepier, sleep-restricted young adults may be less reliant on visual feedback and more reliant on proprioceptive feedback during memory-guided force production. Considering that visual tracking and attention are impaired after sleep restriction (Lim & Dinges, 2010; Heaton et al, 2014), it is possible that sleepier sleep-restricted young adults may trigger a compensatory feedforward strategy while completing the memory-guided force production task, which is more cognitively taxing than the visually guided task (Lim & Dinges, 2010; Heaton et al, 2014). Higher sleepiness could induce a compensatory strategy to mitigate a larger perceived decrement in their force production when visual feedback is removed due to reliance on proprioceptive feedback (Mathew et al, 2021).”

The reviewer is correct that we included all datapoints in the model (please see detailed explanation in response to point #1). Since the model aggregates the data over any not-included factors (trial number and time in seconds), the visualization of the force magnitudes in Figure 3A are reflective of the way the model handles the data: by creating one summarized datapoint per person per day per vision condition.

3. The justification for use of a “mixed effects approach” (line 186) is confusing because it makes it seem like there is a direct association between variability of motor output and model variance, although the association is not so direct. Further, the approach that ends up being used here still has the same requirements, and homoscedasticity is never assessed. Overall, I think that more conventional statistics (perhaps using data transformation if necessary) would make the results much easier to interpret.

Author Response:

We thank the reviewer for raising these concerns. We have rewritten the statistical methods to improve clarity. 

The first paragraph of the statistical methods section now reads:

Line 184-188: “Therefore, the analysis design was repeated measures with two main design factors: day and vision condition. We used a mixed effects multilevel approach to analyze the mean effects of sleep restriction and visual feedback on motor force production, in addition to modeling how the effects of day and vision condition varied across individuals.”

The second paragraph in the results section now reads:

Line 234-237: “The model including random slopes for the day and vision condition fit the data better than the model assuming no variation due to day and vision across individuals. Therefore, random slopes across repeated measures factors were included in the final model.”

Finally, we did not formally test for homogeneity of variance. However, the residuals of the mixed models allow us to discuss the variance across levels of repeated measures factors:

Line 308-321: “To describe the heterogeneity of variance across levels of the repeated measures factors, we investigated the model residuals across levels of day and vision condition, as shown in Figure 5 and described in Table 4. Visual inspection showed that there was larger variability in the residuals (as measured through standard deviation) on the sleep restriction day than at baseline and after recovery sleep, and there was larger variability in residuals during memory-guided than during visually guided force production.”

Minor:

1. My impression is that the association between working memory and memory-guided force production is usually related to how much force production changes after visual feedback is removed (e.g. Vaillancourt & Russell 2002, EBR). Is there a reason this measure was not assessed? This seems potentially important if some of the significance of the study is related to working memory.

Author Response:

Indeed, the paradigm investigated here is that of Vaillancourt and Russell (2002, EBR), in which a finding of “force decay” was revealed when visual feedback was removed. In previous work, we evaluated the presence (or absence) of decay during this “no vision” period (i.e., Neely et al., 2019; Neely, Samimy, et al., 2017; Neely, Wang, et al., 2017). The focus of those four papers (Vaillancourt & Russell, 2022, Neely et al., 2019; Neely, Samimy, et al., 2017; Neely, Wang, et al., 2017) was a between-group analysis to examine how force decay differs in clinical populations. 

In the current work, we utilized the Vaillancourt and Russell (2002) paradigm to evaluate the between- and within- subjects effects of sleep restriction on both visually and memory guided force control. Our examination of force output in these conditions revealed large between-subject variability. In other words, we observed interindividual differences in the response to sleep restriction in both vision conditions. Therefore, the goal of the manuscript was not to examine force decay.

Reviewer 2 suggested we add to the limitations section how the findings “can be generalizable in other force production tasks that include…force production trajectories (e.g., continuous control of force level).” Therefore, the limitations section now includes the following:

Line 407-413: “Also, we modeled force data without including a longitudinal effect (over the 12 s of gripping). As this was the first study exploring the effect of sleep restriction on grip force output, we were interested in the overall effect (or lack thereof) of sleep restriction and recovery sleep on motor output. Future studies may model continuous force production over time to determine if the effects of sleep restriction we found here change with duration of force production.”

REVIEWER #2: 

The authors showed that sleep restriction negatively affected accuracy and variability of force production. One night of recovery sleep was effective to return to the baseline performance only for the visual-guided condition. Although the study results were interesting, this reviewer has concerns, including the details of statistical analysis and how those were reported, and the study rationale.

Author Response:

We thank the reviewer for their helpful comments and suggestions to enhance our manuscript. We have addressed each individual comment below. However, before doing so, we would like to mention two major revisions in response to reviewer feedback. First, we normalized force output to each participant’s maximum voluntary contraction (MVC) before entering force output into the statistical model. As a result, MVC has been removed from the model. We believe this change helps to clarify the Results. This change in our analysis affected the Results, such that the relationship between sleepiness and visual condition depended on the day; higher self-reported sleepiness after sleep restriction and recovery sleep led to smaller memory-guided force.

- # 13: The author mentioned that sleep restriction negatively affects working memory. However, there was a lack of explanations about what working memory is and why investigating the effects of sleep restriction on working memory is essential. Please provide more explanation and supporting evidence that supports your study rationale.

Author Response:

We thank the reviewer for this suggestion. In response, we provide additional background literature as well as clear rationale for studying the effect of sleep restriction on memory-based motor performance.

Line 12-20: “Performance in more complex domains, such as working memory, is also impaired after sleep restriction (Mathew et al, 2021; Drake et al, 2001; Santisteban et al, 2019). Working memory is the temporary storage and manipulation of information, and is an intersection between memory, perception, and the attentional control of behavior (Baddeley, 1992), and is necessary for motor skill acquisition (Adams, 1971; Fitts, 1964). Indeed, the motor system and working memory tasks recruit common neural pathways (Gerver et al., 2020; Marvel, Morgan, & Kronemer, 2019; Neely et al., 2019; Neely, Samimy, et al., 2017; Neely, Wang, et al., 2017). Considering that sleep restriction negatively affects working memory, it follows that sleep restriction may also affect performance of motor tasks that rely on working memory. However, there is a dearth of research examining such tasks.”

- # 17: The participants in this study were young, but the major reference that explains the study rationale comes from a study that examined older adults.

Author Response:

Indeed, one of the seminal papers on memory-guided force output examined older adults with and without Parkinson’s disease (Vaillancourt et al, 2001). More recently, our team has published work examining memory-guided force in individuals ranging from age 7 to 85 (Neely & Chennavasin et al, 2016; Neely & Mohanty et al, 2016; Neely et al, 2017). Additionally, impairments in working memory after sleep restriction have been reported in younger adults (Drake et al., 2001; Mathew et al., 2021; Santisteban, Brown, Ouimet, & Gruber, 2019).

We have added a review reference that encompasses the broad range of literature supporting the overlap between motor system pathways and working memory:

Line 16-17:“Indeed, the motor system and working memory tasks recruit common neural pathways (Gerver et al., 2020; Marvel, Morgan, & Kronemer, 2019; Neely et al., 2019; Neely, Samimy, et al., 2017; Neely, Wang, et al., 2017)."

- # 25 – 31: Many daily living tasks especially involving manual dexterity (e.g., hands) require continuous modulation of force output, but the experimental task used in this study was not the case. Please justify how the experimental task used in this study represents daily motor tasks.

Author Response:

Thank you for indicating the importance of examining how our task translates to real-world tasks. We revised the real-life examples to be more reflective of the task used in this study, which required participants to moderate continuous grip force for 12 seconds. The sentence now reads:

Line 28: “Furthermore, many occupational tasks and activities of daily living require relatively short periods of continuous motor output, such as carrying a cup of hot coffee.”

- #207: The author stated that analyses of variance with Satterthwaite’s method was used. More detail is needed to explain the aim of using the method, which could benefit readers who are not familiar with the method.

Author Response:

This Satterthwaite method of estimating degrees of freedom for a two-sample t-test is used when only the estimates of the variance of distributions are known, as in mixed models. We updated the sentence to include the citation for the package used in R that estimates the degrees of freedom for ANOVAs within mixed models using Satterthwaite’s method. We also cited Satterthwaite’s original paper on this method. The sentence now reads:

Line 196-199: “Within the best-fitting model, analyses of variance with Satterthwaite's method of determining degrees of freedom were used to determine if the interactions and main effects were significant (Bates, 2015; Satterthwaite, 1946), where a priori significance for fixed effects was set at 0.05.”

- #215: It seemed that there were three main factors, but the statistical section would be improved by having more details about what final models the author used and how the results will be interpreted in the Results section.

Author Response:

We added statements to the methods and results sections to better clarify which model was chosen as the final model that was subsequently evaluated for interactions and main effects of the three main factors, as the reviewer stated. 

In the statistical methods section, we added the statement:

Line 215-217: “The best-fitting model of those estimated above, determined by lowest AIC (Lohse, 2020), would be deemed the final model, and the main effects and interactions of this final model would be evaluated.”

In the results section, we added the statement:

Line 234-237: “The model including random slopes for the day and vision condition fit the data better than the model assuming no variation due to day and vision across individuals. Therefore, random slopes across repeated measures factors were included in the final model.”

Results

- Readers could better understand when more direct explanations for the statistical results were included. The authors utilized a mixed model to examine the main and interaction effects with coefficient values in Table 2. However, it would be challenging to interpret the entire table and then match the results in the table with what the authors described their findings in the results section (also for the Discussion). For example, the authors may first describe the best model and the results used to describe the mean differences with statistical results (e.g. p values) in the Results (e.g., Figure 3).

Author Response:

Thank you for the suggestion to improve clarity. We have revised the results section to include statistical results so that readers can follow the statistics without referring to the table. As an example:

Line 277-279: “As shown in Figure 3A, there was an effect of vision condition such that participants produced less force in the memory-guided condition than in the visually guided condition at baseline (Β = -0.711%, p = 0.027).”

- #250: Please provide numbers (“these effects were fairly small in magnitude”).

Author Response:

During manuscript revision, this sentence was removed.

- Table 2: It was challenging to understand this table. As mentioned for the statistical section, there were three main factors (please correct me if I am wrong), and I expected some information about significant interaction between Day (three levels; Baseline, Restriction, and Recovery) and Vision (two levels; Visually-Guided and Memory-Guided). The author should provide more details about how Table 2 can be interpreted.

Author Response:

We thank the reviewer for this suggestion. The interactions between Day and Vision condition are presented in the 6th and 7th lines of Table 2.

In addition to the revisions to the Results section noted above, we revised both the table and the footnote of table 2: 

The footnote now reads: 

Line 263-264: “Note. SE = standard error; VG = Visually Guided; MG = Memory-Guided; KSS = Karolinska Sleepiness Scale. ***p < 0.001; ** p < 0.01; *p < 0.05.”

- When an interaction effect is significant, pair-wise comparisons are usually the next step, as shown in previous studies using ANOVA, but those statistical results with p values were not included. Did not the mixed model provide that information? A similar question was raised in Figure 3 also. Statistical results from within-group comparisons only seemed to be shown, so please provide statistical results for the other paired comparisons (e.g., Visually vs. Memory in the baseline).

Author Response:

The pairwise comparisons are now shown in Table 4 of the estimated marginal means and trends, and indicated in Figures 3 and 4.

- # 285: The authors interpreted the main effect of vision condition to describe their finding, but there was no further information on it. Adding a supporting statistical result can strengthen the sentence.

Author Response:

We have updated the results section to include statistics when results are mentioned. As an example:

Line 277-279: “As shown in Figure 3A, there was an effect of vision condition such that participants produced less force in the memory-guided condition than in the visually guided condition at baseline (Β = -0.711%, p = 0.027).”

- Figure 5: Most results were reported by force relative to MVC (% MVC), but in Figure 5 the data were in newtons. Please clarify what values (e.g., N or % MVC) were used in that Figure. Could the normalization affect the findings?

Author Response:

The analysis has been updated so that the outcome measure is force as a percent of MVC instead of force in newtons. As a result, MVC was removed as an interaction term from the model. Therefore, all figures and analyses (including the figure indicated in this comment figure 5) are relative to force as a percent of MVC.

Discussion

- The study findings were from an isometric pinching force production at a specific constant force output (25% MVC). The authors needed to address how the findings can be generalizable in other force production tasks that include different force production types (e.g., concentric), different amounts of force levels (less or higher than 25% MVC), and force production trajectories (e.g., continuous control of force level). Adding that information in a separate study limitation section can address this issue.

Author Response:

The reviewer brings up a good point. We have added a limitations section before the conclusions paragraph, copied below:

Line 400-413: “This study had some limitations. First, the force task asked participants to maintain a grip force output of 25% of MVC. While this task is relatable to some activities of daily living, such as holding hot beverages, it may not be directly applicable to other activities requiring changes in grip force (such as operating heavy machinery) or gripping at higher or lower levels of force (such as holding a pencil or a large child). Future studies should investigate the effect of sleep restriction on various motor force tasks to better understand the wholistic effect of sleep restriction on movement. Also, we modeled force data without including a longitudinal effect (over the 12 s of gripping). As this was the first study exploring the effect of sleep restriction on grip force output, we were interested in the overall effect (or lack thereof) of sleep restriction and recovery sleep on motor output. Future studies may model continuous force production over time to determine if the effects of sleep restriction we found here change with duration of force production.”

References

Gelman, A., and Hill, J. (2008). Data Analysis Using Regression and Multilevel/Hierarchical Models. Cambridge, MA: Cambridge University Press.

Bates D. Fitting linear mixed-effects models using lme4. Journal of Statistical Software. 2015;67(1):1-48.

Neely, K. A., Chennavasin, A. P., Yoder, A., Williams, G. K., Loken, E., & Huang-Pollock, C. L. (2016). Memory-guided force output is associated with self-reported ADHD symptoms in young adults. Exp Brain Res, 234(11), 3203-3212. doi:10.1007/s00221-016-4718-1

Neely, K. A., Mohanty, S., Schmitt, L. M., Wang, Z., Sweeney, J. A., & Mosconi, M. W. (2016). Motor Memory Deficits Contribute to Motor Impairments in Autism Spectrum Disorder. J Autism Dev Disord. doi:10.1007/s10803-016-2806-5

Neely, K. A., Samimy, S., Blouch, S. L., Wang, P., Chennavasin, A., Diaz, M. T., & Dennis, N. A. (2017). Memory-guided force control in healthy younger and older adults. Experimental brain research, 235(8), 2473-2482.

Marvel, C. L., Morgan, O. P., & Kronemer, S. I. (2019). How the motor system integrates with working memory. Neurosci Biobehav Rev, 102, 184-194. doi:10.1016/j.neubiorev.2019.04.017

Lohse KR, Shen J, Kozlowski AJ. Modeling Longitudinal Outcomes: A Contrast of Two Methods. Journal of Motor Learning and Development. 2020;8(1):145-65.

---

## [Decision Letter · Decision Letter 1]

28 Jul 2022

PONE-D-22-02851R1Sleep Restriction Impairs Visually and Memory-Guided Force ControlPLOS ONE

Dear Dr. Brinkerhoff,

Thank you for submitting your manuscript to PLOS ONE. After careful consideration, we feel that it has merit but does not fully meet PLOS ONE’s publication criteria as it currently stands. Therefore, we invite you to submit a revised version of the manuscript that addresses the points raised during the review process.

We look forward to receiving your revised manuscript.

Kind regards,

Kenichi Shibuya, Ph.D.

Academic Editor

PLOS ONE

Additional Editor Comments :

I just received the comments of the reviewers. Two reviewers suggest to me "Accept". But Reviewer 2 suggests "Major Revision".

Reviewer 2 suggests the important issues for improving your manuscript. Please respond to the comments from Reviewer 2.

Reviewers' comments:

Reviewer's Responses to Questions

**Comments to the Author**

1. If the authors have adequately addressed your comments raised in a previous round of review and you feel that this manuscript is now acceptable for publication, you may indicate that here to bypass the “Comments to the Author” section, enter your conflict of interest statement in the “Confidential to Editor” section, and submit your "Accept" recommendation.

Reviewer #1: All comments have been addressed

Reviewer #2: (No Response)

Reviewer #3: All comments have been addressed

2. Is the manuscript technically sound, and do the data support the conclusions?

Reviewer #1: Yes

Reviewer #2: Partly

Reviewer #3: Yes

3. Has the statistical analysis been performed appropriately and rigorously? 

Reviewer #1: I Don't Know

Reviewer #2: I Don't Know

Reviewer #3: Yes

4. Have the authors made all data underlying the findings in their manuscript fully available?

Reviewer #1: Yes

Reviewer #2: Yes

Reviewer #3: Yes

5. Is the manuscript presented in an intelligible fashion and written in standard English?

Reviewer #1: Yes

Reviewer #2: Yes

Reviewer #3: (No Response)

6. Review Comments to the Author

Reviewer #1: Overall I believe the authors have satisfactorily addressed my concerns. I am still unsure of the applicability of the statistical methodology (in particular treating individual timepoints as samples) and hope a domain expert can give further feedback here. I don't expect results will change if statistics were carried out differently.

Reviewer #2: The authors responded to most of the comments raised by the two reviewers well, but it still needs details and clarification about how they performed the statistical analysis and reported the results. The authors should provide more details about how the force data were analyzed to perform the statistical analysis.

Methods

#179-181: The authors did not fully explain the normalization procedure using the MVC values and how the force data were analyzed.

#183: Please provide readers with details for the levels of each factor sooner than later (e.g., two levels, no visual feedback, and visual feedback). Readers need to wait until seeing that information in lines 203~204.

#187: Please provide readers with details for what variables were used as dependent variables in the analysis (e.g., How was the performance of motor force production evaluated? mean of force data normalized by MVC values for each day?)

Results

#234: Please provide a quantitative information that explains the model selection (e.g., AIC values).

# Table 2. The authors stated that there are two factors in the Method, but there seemed to be another factor (KSS; e.g., #282). It is confusing which one is right. Could the authors clarify this issue?

Discussion

#402-403: Please provide references to support that the force level used in the current study matches the force levels required for performing some activities of daily living.

Reviewer #3: The statistical analysis of the paper is well done and described. I have no comments for the authors and I suggest accepting the paper in its current version.

7. PLOS authors have the option to publish the peer review history of their article (what does this mean?). If published, this will include your full peer review and any attached files.

Reviewer #1: No

Reviewer #2: No

Reviewer #3: No

---

## [Author Response · Author response to Decision Letter 1]

15 Aug 2022

REVIEWER #1: 

The authors responded to most of the comments raised by the two reviewers well, but it still needs details and clarification about how they performed the statistical analysis and reported the results. The authors should provide more details about how the force data were analyzed to perform the statistical analysis.

Methods

#179-181: The authors did not fully explain the normalization procedure using the MVC values and how the force data were analyzed.

Author Response:

We have updated the end of the “Motor Force task” section to include information about how the force data were normalized to MVC.

-Line 181-183: “Force data were collected in newtons and were divided by the participant’s MVC measured on the same day, multiplied by 100%. Therefore, the data were analyzed as a percent of MVC.”

#183: Please provide readers with details for the levels of each factor sooner than later (e.g., two levels, no visual feedback, and visual feedback). Readers need to wait until seeing that information in lines 203~204.

Author Response:

We named the factor levels earlier in the statistical analysis section, where the reviewer indicated. We also named the explored covariates in the same sentence.

-Line 186-189: “Therefore, the analysis design was repeated measures with two main design factors — day (baseline, sleep restriction, recovery sleep) and vision condition (visually guided and memory-guided) — and four potential covariates (race, age, TST, and KSS score).”

#187: Please provide readers with details for what variables were used as dependent variables in the analysis (e.g., How was the performance of motor force production evaluated? mean of force data normalized by MVC values for each day?)

Author Response:

We provided detailed on the exact dependent measure used in the statistical analyses.

-Line 190-193: “We used a mixed effects multilevel approach to analyze the effects of sleep restriction and visual feedback on mean force produced in the last 12 seconds of each trial, normalized by MVC, in addition to modeling how the effects of day and vision condition varied across individuals.”

-Line 198-200: “A series of models were estimated in a two-level multilevel framework using the lme4 package [36] in R [37] to model the mean force, normalized by MVC, across days and vision condition, nested within participants.”

Results

#234: Please provide a quantitative information that explains the model selection (e.g., AIC values).

Author Response:

We added AIC fits for the final model (allowing variation due to levels of repeated measures) and for the model that assumed no variation due to repeated measures factors.

-Line 186-189: “The model including random slopes for the day and vision condition (AIC = 1.06x10^6) fit the data better than the model assuming no variation due to day and vision across individuals (AIC = 1.08x10^6).

In total, 16 models were built-up and consecutively compared for fit to the data. The authors feel that listing all model fits would take space and detract from the results; therefore, all other AIC values (and other model fit values) are provided in supplementary tables.

# Table 2. The authors stated that there are two factors in the Method, but there seemed to be another factor (KSS; e.g., #282). It is confusing which one is right. Could the authors clarify this issue?

Author Response:

While there were two design factors, we also included any covariate that significantly improved the model — KSS score was the only covariate that improved model fit to the data. This has been clarified in manuscript in the following lines.

-Line 186-189: “Therefore, the analysis design was repeated measures with two main design factors — day (baseline, sleep restriction, recovery sleep) and vision condition (visually guided and memory-guided) — and four potential covariates (race, age, TST, and KSS score).”

-Line 244-246: “The best-fitting model, shown in Table 2, included main effects for day, vision condition, KSS score, and the three-way interaction among day, vision condition, and KSS score.”

Discussion

#402-403: Please provide references to support that the force level used in the current study matches the force levels required for performing some activities of daily living.

Author Response:

This sentence has been revised to be more general, and citations have been added.

-Line 402-406: “Grip force of 20-40% MVC is important for activities of daily living such as writing, unlocking a door, and tying shoes (Marshall & Armstrong, 2004), but it may not be directly applicable to other activities requiring higher or lower grip force (such as operating heavy machinery).”

References

Marshall, M. M., & Armstrong, T. J. (2004). Observational Assessment of Forceful Exertion and the Perceived Force Demands of Daily Activities. Journal of Occupational Rehabilitation, 14(4), 281–294. https://doi.org/10.1023/B:JOOR.0000047430.22740.57

---

## [Decision Letter · Decision Letter 2]

23 Aug 2022

Sleep Restriction Impairs Visually and Memory-Guided Force Control

PONE-D-22-02851R2

Dear Dr. Brinkerhoff,

We’re pleased to inform you that your manuscript has been judged scientifically suitable for publication and will be formally accepted for publication once it meets all outstanding technical requirements.

Kind regards,

Kenichi Shibuya, Ph.D.

Academic Editor

PLOS ONE

Additional Editor Comments (optional):

Reviewers' comments:

Reviewer's Responses to Questions

**Comments to the Author**

1. If the authors have adequately addressed your comments raised in a previous round of review and you feel that this manuscript is now acceptable for publication, you may indicate that here to bypass the “Comments to the Author” section, enter your conflict of interest statement in the “Confidential to Editor” section, and submit your "Accept" recommendation.

Reviewer #2: All comments have been addressed

2. Is the manuscript technically sound, and do the data support the conclusions?

Reviewer #2: Yes

3. Has the statistical analysis been performed appropriately and rigorously? 

Reviewer #2: Yes

4. Have the authors made all data underlying the findings in their manuscript fully available?

Reviewer #2: Yes

5. Is the manuscript presented in an intelligible fashion and written in standard English?

Reviewer #2: Yes

6. Review Comments to the Author

Reviewer #2: The authors responded to the comments well. Although I am not an expert in statistics, this reviewer thinks that the details for statistical analysis that the author provided could enable readers to replicate their work, which may result in other follow-ups and exciting research on that topic.

7. PLOS authors have the option to publish the peer review history of their article (what does this mean?). If published, this will include your full peer review and any attached files.

Reviewer #2: No

---

## [Editor Report · Acceptance letter]

25 Aug 2022

PONE-D-22-02851R2 

Sleep Restriction Impairs Visually and Memory-Guided Force Control 

Dear Dr. Brinkerhoff:

I'm pleased to inform you that your manuscript has been deemed suitable for publication in PLOS ONE. Congratulations! Your manuscript is now with our production department. 

Kind regards, 

on behalf of

Dr. Kenichi Shibuya 

Academic Editor

PLOS ONE